# Predictors of Dietary Diversity of Indigenous Food-Producing Households in Rural Fiji

**DOI:** 10.3390/nu11071629

**Published:** 2019-07-17

**Authors:** Lydia O’Meara, Susan L. Williams, David Hickes, Philip Brown

**Affiliations:** 1Department of Medical Sciences, School of Health, Medical and Applied Sciences, CQUniversity, Shield and Abbott Streets, Cairns QLD 4870, Australia; 2Department of Physical Research Group, Appleton Institute, School of Health, Medical and Applied Sciences, CQUniversity, Bruce Highway, Rockhampton QLD 4702, Australia; 3Secretariat of the Pacific Community, Sigatoka Agriculture Research Station, Sigatoka, Fiji; 4Department of Agriculture, Science and the Environment, School of Health, Medical and Applied Sciences, CQUniversity, 6 University Drive, Bundaberg QLD 4670, Australia

**Keywords:** dietary diversity, farm diversity, food security, household, indigenous, agriculture, Fiji

## Abstract

Fiji, like other Pacific Islands, are undergoing economic and nutrition transitions that increase the risk of noncommunicable diseases (NCDs) due to changes of the food supply and dietary intake. This study aimed to examine dietary diversity (DD) in indigenous food-producing households in rural Fiji. Surveys were conducted with households from the *Nadroga-Navosa,*
*Namosi* and *Ba* Provinces of Western Fiji in August 2018. Participants reported on foods consumed in the previous 24 h per the Household Dietary Diversity Score. Data was analysed using multinomial logistic regression. Of the 161 households, most exhibited medium DD (66%; M = 7.8 ± 1.5). Commonly consumed foods included sweets (98%), refined grains (97%) and roots/tubers (94%). The least consumed foods were orange-fleshed fruits (23%) and vegetables (35%), eggs (25%), legumes (32%) and dairy (32%). Households with medium DD were more likely to be unemployed (OR 3.2, *p* = 0.017) but less likely to have ≥6 occupants (OR = 0.4, *p* = 0.024) or purchase food ≥2 times/week (OR = 0.2, *p* = 0.023). Households with low DD were more likely to have low farm diversity (OR = 5.1, *p* = 0.017) or be unemployed (OR = 3.7, *p* = 0.047) but less likely to have ≥6 occupants (OR = 0.1, *p* = 0.001). During nutrition transitions, there is a need for public health initiatives to promote traditional diets high in vegetables, fruits and lean protein and agricultural initiatives to promote farm diversity.

## 1. Introduction

Globally, nutrition-related noncommunicable diseases (NCDs) account for more than 36 million deaths each year and almost 80% of these deaths occur in low- and middle-income countries [1]. Over the last 30 years, the prevalence of NCDs has risen faster in the Pacific region compared to the rest of the world [2,3,4]. In this region, a majority of premature deaths (before 60-years-old) are related to NCDs such as diabetes and cardiovascular disease (CVD) [5,6,7], which has resulted in “a human, social and economic crisis” [8]. Economic transition to a cash-based society can increase the risk of NCDs due to significant changes to the food supply and in turn dietary intake [9,10].

Fiji is an upper-middle-income country located in the South Pacific Ocean with the second largest population (0.9 million) of the Pacific Island countries [11]. Indigenous *iTaukei* Fijians account for over 60% of the population and they have historically lived on a subsistence-based diet of starchy root staples, green leafy vegetables and seafood [11]. Over the last 60 years, traditional food systems have deteriorated due to rapid urbanisation and globalisation [12,13]. This transition from a subsistence to a cash-based society has led to a higher consumption of imported foods, which are low in fibre and nutrients and high in energy [14,15,16]. While dietary quality has fallen, Fiji has experienced a dramatic rise in prevalence of obesity and NCDs and life expectancy has stagnated as a result [5,17].

Dietary diversity, defined as “the number of different foods or food groups consumed over a given reference period” is considered a key component of healthy diets [18] and highly diverse diets are associated with lower risk of micronutrient deficiencies [19], weight gain [20] and NCDs [21,22]. Low dietary diversity is a significant concern for poor and developing countries where diets are based mainly on carbohydrates and include limited amounts of animal products and fresh fruits and vegetables [15,18,23].

Over 77% of poor households in low- and middle-income countries exhibit low dietary diversity that places them at higher risk of negative health outcomes such as obesity and NCDs [24,25]. In general, healthier diets cost more than unhealthy diets [26,27,28,29,30] and it has been argued that the food budgets of low-income populations are insufficient to guarantee access to a nutritious diet, which means that socially disadvantaged groups consume disproportionally high intakes of empty calories to satisfy hunger [30,31,32]. Evidence from studies of rural populations that are involved in food production is equivocal [23,33]. In some studies, farm diversity has been associated with increased dietary diversity of the farming households [34,35]; and in other studies access to markets to sell crops for income has mitigated associations between agricultural land ownership and dietary intake [36,37].

In 2014, over 36% of Fijians lived in poverty, which means they lacked adequate resources to meet the basic needs of the household [38]. The majority of rural Fijians rely on agriculture for income, an albeit volatile source, meaning they account for over 62% of Fiji’s poor [38]. In addition, Fijian farming communities are increasingly vulnerable to the negative impacts of weather shocks such as climate change and cyclones, which limit their ability to sustain food production [39,40,41]. For example, Tropical Cyclone Winston hit Fiji in 2016 and the subsequent destruction of infrastructure and food production reduced the ability of households to consume their own crops and generate income to purchase food [42].

Previous studies have examined dietary diversity in urban Pacific Islander populations [14,17,43,44,45,46]. Despite considerable investment in agriculture development programs by government and international aid donors to improve local food production and nutrition outcomes in rural areas, there are no known studies examining dietary diversity of food-producing communities in rural Fiji. As a first step in developing initiatives to increase dietary diversity, it is important to understand current dietary consumption patterns and the influence of underlying drivers of food intake [18,23,43,47,48]. The aim of this study was to examine associations between household dietary diversity and a range of personal and household characteristics and farm diversity in a sample of indigenous food-producing households in rural Fiji. Findings from this study will contribute information for future development of effective dietary diversity initiatives that aim to improve dietary quality and related health outcomes.

## 2. Materials and Methods

### 2.1. Study Design

This cross-sectional pilot study was part of a larger mixed-methods study examining factors influencing the food security and dietary diversity of indigenous food-producing households in rural Fiji. The research team worked collaboratively with in-country partners throughout the project. This included intensive engagement with a range of stakeholders over approximately nine months prior to commencement of in-country activities. This was considered integral to developing effective partnerships to support conduct of research activities including recruitment of participants, development of study processes, materials, and tools, and collection of data.

### 2.2. Ethical Approval

Ethical approval was obtained from the Fiji National Research Ethics Review Committee (2018.99.WES) and CQUniversity’s Human Research Ethics Committee (HREC 2018-006; approval number 21082). Letters of support were also obtained from in-country stakeholders: (i) Secretariat of the Pacific Community and Fiji Government Ministries of (ii) Health, (iii) Agriculture, (iv) *iTaukei* Affairs and (v) Education.

### 2.3. Participant Recruitment

Eight rural villages were selected from five different *Tikinas* (subunits of a province) from the *Nadroga-Navosa, Namosi* and *Ba* Provinces in Western Fiji. As this was a feasibility study, villages were selected based on convenience, affiliation with agriculture development programs and geographical proximity to the *Sigatoka Valley.* This large area is known locally as the ‘salad bowl’ of Fiji due to high rates of food production. Based on consultation with in-country partners, further villages were selected to represent a range of agriculture and socioeconomic conditions.

The research team met in person with the chief/elders of each village and where applicable the leader of local farming groups, to obtain approval to conduct the study. All eight villages agreed to participate. Information sessions were held at a neutral location within each village (i.e., village meeting hall) with households interested in participating. Participation was voluntary. To be eligible for inclusion, the participant reporting for each household had to be aged 18 years or older and have knowledge of what foods members of the household ate at home in the previous 24-h. Prior to consenting to participate in the study, households were provided with verbal and written information in English and translation into local dialect was conducted as necessary. Percent of households sampled varied between 10–90% of village total dependent on village size and data collection time constraints.

Monetary compensation was not given to any villages or participants; however, village chiefs/elders did request a report of preliminary findings, and practical nutrition education workshops tailored to identified nutrition needs of each village. This method of returning knowledge to the community and including all members of the village in a tangible outcome such as nutritional education is considered appropriate in collectivist-based cultures where inclusivity is an important component of all activities [49].

### 2.4. Survey Administration

Data was collected during the height of the Fijian harvest season from 31 July to 28 August 2018. The research team (consisting of the lead researcher, indigenous *iTaukei* Fijian co-researcher, a research assistant, and local Ministry of Agriculture extension officers) visited villages as a group. Farming group managers and village health workers also volunteered their assistance with translation and data collection. The research team sought guidance from each village headman before entering a village. The survey consisted of three sections: (i) personal and household characteristics, (ii) farm diversity, and (iii) household dietary diversity. Data was collected using the CommCare mobile application (Dimagi Inc., Cambridge, MA, USA, 2018).

### 2.5. Personal and Household Characteristics

Participants reported their personal characteristics including: age (years) (from: 18–29, 30–54, ≥ 55 (dichotomised 18–54, ≥ 55); gender; ethnicity; living location (village); education attainment (≤12 (did not complete secondary school), ≥13 (completed secondary school or higher) years of education); employment status (unemployed (caregiver, community/-religious commitments), employed (works on family farm, other paid employment)); and number of chronic health conditions (from: arthritis, asthma, back/-neck pain, cancer, depression/anxiety, diabetes, heart disease, hypertension, kidney disease, stroke) (see Table 1).

Participants also reported household characteristics including: gross annual household income (FJ$) (from: ≤1000, 1001–5000, 5001–15,000, 15,001–25,000, ≥25,001, unknown; dichotomised ≤5000, ≥5001); primary source of household income (from: self-employed farm, other [includes other small business], private sector, public sector, remittance); number of household occupants (1–5, ≥6); number of children aged 0–5-years in household (0–2, ≥3); food purchase frequency (≥2, ≤1 per week); farm status (from: subsistence, semi-commercial, commercial); and number of livestock and crop species grown/available from natural resources. In *iTaukei* culture, age and income are sensitive questions, therefore data was collected using categories to respect privacy of participants. Furthermore, monetary expenditure can also be a sensitive question, therefore food purchase frequency was used as a proxy for food acquired from markets in comparison to food acquired from own production [50]. Socioeconomic data (i.e., age, household occupancy, income) were defined and categorised according to standard Fijian census data [11,51] and in line with dichotomised cut-offs for Fijian data utilised by the World Bank [38] and World Vegetable Center [52] to enable comparison.

### 2.6. Farm Diversity

In this study, we used the Household Biodiversity Index (HBI) as a proxy for farm diversity that has been validated for use in agriculture-dependent communities [35,36,53]. The HBI is a simple count of all crops and livestock available to the household from the farm or nearby natural resources. Each crop or livestock species is given a quantitative value of one and expressed as a Crop- or Livestock Biodiversity Index [35]. Because *iTaukei* have access to fruits from communal trees and green leafy vegetables that grow wild, crops reportedly gathered by hand were included in the Crop Biodiversity Index. Due to availability of fish/seafood from local rivers and oceans and wild pigs from the forest, livestock that was reported to be fished, hunted or gleaned from the nearby environment was also included in the Livestock Biodiversity Index. Crop and Livestock Biodiversity Indexes were summed to produce a continuous HBI for each household. Based on the sample mean, farm diversity scores were then dichotomised into low (1–7) or high (8–28).

### 2.7. Household Dietary Diversity

Household dietary diversity was collected using the United Nation Food and Agriculture Organization’s (FAO) Household Dietary Diversity Score (HDDS) [54]. The HDDS reflects nutrient adequacy of household diets and is positively associated with adequate micronutrient intake at all stages of life [54,55,56]. It is recommended for assessing changes in dietary diversity of populations living in agriculture-dependent areas; is validated for use in a range of low- and middle-income countries; and is low-cost, quick, and easily administered [54].

The reference period of foods consumed by the household in the previous 24-h was selected for use in this study. If the household had participated in a cultural function (i.e., feast) in the last 24-h then the reference period of the day before the function was used. The use of the 24-h reference period for food intake was chosen because it is less cumbersome for the respondents, is subject to less recall error and aligns to the recall periods used in many other studies [55,56,57,58,59].

Household food consumption was collected in accordance with FAO Guidelines for Measuring Household and Individual Dietary Diversity (further information available at www.fao.org) [54]. Questions were adjusted to use culturally appropriate food names and examples following testing with Fijian stakeholders (see Table 2 for local food examples). Starting with the first meal or beverage in the morning, participants were asked to describe the foods that the household ate or drank the day before at home. When composite dishes were mentioned, enumerators probed for a list of ingredients. When the respondent finished, enumerators also probed for meals or snacks that may have been overlooked. Ingredients eaten at home were recorded under the corresponding food groups using a list-based method (see Table 2 for food groups) [54]. As per FAO guidelines, it was noted if household members ate outside of the home in the recall period; however, ingredients eaten outside of the home were not recorded. The HDDS food consumption data was categorised into seven food groups per the World Health Organization’s (WHO) Minimum Acceptable Diet (MAD) [60]: (i) carbohydrates (grains, roots and tubers), (ii) legumes and nuts, (iii) dairy, (iv) flesh meat, (v) eggs, (vi) vitamin-A rich fruits and vegetables, and (vii) other fruits and vegetables (see Table 3). MAD scores were categorised according to WHO prescribed cut-offs: (i) low (1–3), (ii) medium (4–5), and (iii) high (6–7) [60].

### 2.8. Statistical Analysis

Analyses were performed using IBM SPSS (v25.0, IBM Corporation, Armonk, NY, USA). Descriptive statistics were used to present socioeconomic characteristics of the sample, reporting mean and standard deviation for continuous data and percentages for categorical variables. Multinomial logistic regression models were used to examine the relationship between personal and household characteristics, farm diversity and MAD scores (reference category ‘high’). Univariate analysis was conducted with all independent variables and all variables with significant associations (employment, household occupants, food purchase and farm diversity) were retained in the final adjusted model. Confidence intervals of 95% and a *p*-value of <0.05 were assumed for statistical significance.

## 3. Results

### 3.1. Descriptive Statistics of Respondents and Households

A total of 161 households from eight villages of Western Fiji’s *Nadroga-Navosa, Namosi* and *Ba* Provinces were included in the analysis (see Table 1 for descriptive statistics). All respondents were indigenous *iTaukei*. Majority were female (73%), with no secondary education (71%), unemployed (68%) and aged 18–54-years (71%). The most commonly reported chronic conditions were back/-neck pain (35%), high blood pressure (28%), arthritis (11%), and diabetes (10%).

Most households had 1–5 occupants (61%), ≥3 children aged 0–5-years (58%), and lived on low annual household incomes (FJ$ 1000–15,000, 64%). Majority of households purchased food ≥2 times per week (78%).

Most were self-employed smallholder farming households (77%) running semi-commercial farms (subsistence farmers that sell surplus food crops for income) (73%). The majority of households had low farm diversity (62%) and the mean farm diversity index was 7.9 ± 5.2. Majority had low crop biodiversity index (mean 7.1 ± 5.1, 68%) and low livestock biodiversity index (mean 0.9 ± 1.2, 54%).

Over 85% of households had low or medium HDDS (19% and 67% respectively). The mean HDDS was 7.8 ± 1.5 (between 0–12). Furthermore, the mean MAD score was 4.3 ± 1.2 (between 0–7). Approximately 75% of households consumed the minimum number of 4 MAD food groups as recommended by the WHO.

### 3.2. Household Dietary Diversity and Farm Diversity by Food Group

In the 24-h period before the survey, the most commonly consumed food groups were high-sugar food/drink (98%), refined grains (97%), white roots/tubers (94%), flavourings/other drinks (91%), oils/fats (82%), and dark green leafy vegetables (77%) (see Table 2 for frequency of household dietary diversity and farm diversity by food groups). Least consumed food groups were orange-fleshed fruits (23%) and vegetables (35%), eggs (25%), dried legumes/nuts (32%), and dairy products (32%). Each household also reported on food crops grown, livestock owned, or foods sourced from the natural food environment. The most commonly grown/-foraged food groups were: white roots/tubers (97%), other vegetables (68%), dark green leafy vegetables (65%), fish/seafood (37%) and other fruits (34%).

### 3.3. Percentage of Food Groups Consumed by Household Dietary Diversity Tertile

A similar dietary pattern was observed for all WHO MAD tertiles with percent differences observed between the low and high tertiles (see Table 3 for percent of food groups consumed by households in the 24-h food recall period by household dietary diversity tertile). The most commonly consumed food groups for all tertiles (low, medium and high) were carbohydrates (100% for all), flesh meat (63%, 78%, and 97% respectively) and vitamin-A rich fruits and vegetables (53%, 95%, and 97%). For all tertiles, the least frequently consumed food groups were other fruits and vegetables (34%, 93% and 97% respectively), legumes (8%, 29%, and 77%), eggs (8%, 26%, and 77), and dairy (8%, 14%, and 71%).

### 3.4. Univariate Associations between Variables and Dietary Diversity

Households with low dietary diversity were more likely than those with high dietary diversity to be unemployed (OR = 7.4; *p* = 0.001) or have low farm diversity (OR = 9.1; *p* = 0.000) but less likely than households with ≥6 or more occupants (OR = 0.1; *p* = 0.000) (see Table 4 for univariate logistic regression analysis). Households with medium dietary diversity were more likely to be unemployed (OR = 3.3, *p* = 0.005) while households with ≥6 household occupants (OR = 0.4; *p* = 0.044) or who purchased food ≥2 times per week (OR = 0.3; *p* = 0.045) were more likely to have high dietary diversity. Neither gender, age, education, income, or number of children 0–5-years-old was significantly associated with household dietary diversity.

### 3.5. Predictors of Household Dietary Diversity

After controlling for all other significant factors and compared to those with high dietary diversity, households with low dietary diversity were more likely to be unemployed (OR = 3.7; *p* = 0.047) or have low farm diversity (OR = 5.1, *p* = 0.017) while households with ≥6 household occupants (OR = 0.1, *p* = 0.001) were more likely to have high dietary diversity (see Table 5 for adjusted multinomial logistic regression analysis).

After controlling for all other significant factors and compared to those with high dietary diversity, households with medium dietary diversity were more likely to be unemployed (OR = 3.2, *p* = 0.017) while households with ≥6 household occupants (OR = 0.4; *p* = 0.024) or who purchased food ≥2 times per week (OR = 0.2; *p* = 0.023) were more likely to have high dietary diversity.

## 4. Discussion

To our knowledge, this study is the first to examine predictors of household dietary diversity in indigenous food-producing communities in Fiji. In this sample, we observed a dietary pattern that included a combination of both traditional and imported foods. This suggests that rural populations are undergoing the nutrition transition to a ‘modern diet’ at a slower rate in comparison with urban populations [15,43,61]. Households were at higher risk of experiencing low dietary diversity if the household had low farm diversity or if the survey respondent was unemployed. In contrast, living in a household with more than six occupants or purchasing food twice or more per week was a predictor of higher dietary diversity.

Majority of households (75%) in this sample consumed the minimum number of four different nutrient-rich food groups recommended by the WHO to ensure optimal micronutrient intake. This high dietary quality contrasts to studies conducted in the remote Papua New Guinean (PNG) highlands and small island atoll of Kiribati, where 60% of participants did not consume the minimum number of food groups [44,62]. The mean HDDS of our study was 7.8, which was higher than the mean DDS of approximately 3.0 in PNG [62], 4.0 in Kiribati [44], and 6.5 [45] and 7.3 [46] in the Solomon Islands. Except for the PNG study, all of these samples were urban populations, which suggests that rural food-producing communities in Fiji have higher dietary diversity. It is important to note that diet quality in farming communities fluctuate with food production cycles [33,63]. This feasibility study was undertaken during the peak of the Fijian harvest season, which may account for the high prevalence of micronutrient adequacy. Further research is required to determine seasonal differences in diet diversity that may be lower during the off-season [33,63].

A majority of households (73%) in this sample were subsistence farmers who sold surplus food crops for income. Farm diversity (range of livestock and crop species grown) can help households access foods either directly through consumption of food from own farm production or indirectly through income generated from sale of farm produce [63]. It follows that the households that were involved in paid employment or who had higher farm diversity were more likely to have higher dietary diversity. At the national level, only 10–17% of Fijians consume the main traditional food crops of starchy root staples (i.e., *cassava* and *taro*) and dark green leafy vegetables (i.e., *bele* and *rourou)* daily [64,65]. Although approximately 35% of Fijians are predominately Indo-Fijian with different cultural diets to indigenous *iTaukei*, this percentage is still low. This is partly due to difficulties urban Fijians face in sourcing sufficient fresh fruits and vegetables that are affordable [66,67,68]. Many of the households in our sample grew white roots or tubers (97%), other vegetables (68%) and dark green leafy vegetables (65%). In turn, a majority of this sample also ate root crops (97%) and green leafy vegetables (77%) highlighting the positive impact access to land and subsistence food production imparts on dietary intakes. Therefore, the agriculture initiatives in this area should promote farm diversity as Fiji continues the nutrition transition trajectory.

In this study, living in a household with six or more occupants was a predictor of higher dietary diversity. Fijian households are 50% more likely to be poor if they have large households with multiple dependents (children and elderly) [51], or if they are of indigenous *iTaukei* ethnicity and self-employed food-producers with low education attainment, which is representative of our sample [38]. Examination of data from this study sample showed that the larger households were indeed poorer in relation to the smaller households. Globally, large households with low incomes often have lower dietary diversity due to the high cost of nutrient-dense foods [27,30,31]. However, the large households in this study had the ability to supplement diet regardless of income due to ability to grow food and source seafood from the South Pacific Ocean or *Sigatoka* and *Ba* Rivers. Although poverty may have lowered their purchasing power to buy store-bought foods, it has forced them into a more traditional subsistence-based existence, which has increased their diet quality. This finding demonstrates how important access to land and fishing grounds is to enable dietary diversity of poor households in rural Fiji.

Households in this sample were more likely to have medium dietary diversity if they purchased food (i.e., groceries) for the household two or more times per week. Households can rarely meet all dietary needs from their own farm’s production (especially during the off-season); therefore purchasing food can increase dietary diversity [63]. Conversely, access to store-bought foods increases access to unhealthy foods, which are often cheaper in comparison with nutrient-rich foods [30]. In Fiji, unhealthy food items are indeed cheaper due to national price control measures on sugar and white flour [69]. Furthermore, processed food items are widely available in Fiji due to dependence on imports [67], which correlates with increases in NCDs [17]. Therefore, this finding should be interpreted with caution.

Over 97% of households in this sample consumed more foods from the empty-calorie food groups (sugar-sweetened beverages and foods, and white processed grains) in comparison to the nutrient-rich groups. Notably, the HDDS is a method of counting food groups that does not account for excess energy intake; therefore, a high score is not synonymous with a good quality diet [60]. Indeed, the high consumption of empty-calorie foods identified in this sample is similar to those observed elsewhere in Fiji demonstrating that the nutrition transition is underway in rural areas. For example, 90% of Fijian school-aged children drink sugar-sweetened beverages on a regular basis [43]. At the national level and in parallel with increased intake of empty-calorie foods, risk factors for NCDs such as obesity are on the rise with up to 50% of indigenous Fijian women now obese (BMI ≥ 30 kg/m^2^) [70,71]. To reduce the risk of obesity and NCDs in these communities, explicit nutrition education is needed to lower intake of sugar and processed foods [17].

It is well established that increased consumption of vegetables, fruits and legumes is associated with lower risk of obesity and diabetes [72] and high intake of fruits and vegetables helps prevent cancer and CVDs [73]. Despite being food producers, at least 35% of households in this study did not grow green leafy vegetables and 60% did not grow orange-fleshed fruits or vegetables. Many of the agriculture initiatives in Fiji are established on principles of a cash-based economy that accentuates sale of food crops for income [74,75]. Concerningly, crops intended for commercial sale have displaced the planting of traditional foods for home consumption [39,76]. Dark green leafy vegetables are important sources of iron, vitamin A, and vitamin C in Fijian diets [77]. Orange-fleshed fruits and vegetables, legumes, eggs and dairy are also important sources of these essential micronutrients [77]. This demonstrates that between 35–65% of the households in this study could benefit from increasing production and consumption of these nutrient-rich food groups.

Furthermore, over half of households (58%) in this study had three or more children under 5 years-old. At the national level, 74% of Fijian school-aged children do not eat the WHO recommended 5+ servings of fruits and vegetables per day, similar to developed nations [43]. In turn, iron-deficiency anaemia has increased markedly in the general Fijian population over the last 10-years and is a significant public health concern. Specifically, 62% of Fijian children under-5-years-old have anaemia and 48% of adolescents up to 17 years-old demonstrating that diets do not contain adequate iron required for physical and cognitive growth [64,65]. Vitamin-A deficiency is also on the rise with up to 18% of the general Fijian population now deficient [64,65]. Thus, to ensure equitable health outcomes for the next generation, it is important that dietary diversity initiatives focus on increasing intake of nutrient-rich foods in rural households with young children.

Over 70% of survey respondents in this study had not completed secondary school. Both longitudinal and cross-sectional studies highlight that low education is associated with low dietary diversity [78,79]. Nutritional knowledge is positively associated with higher quality diets [80,81,82]. However, low education can limit the ability of individuals to acquire nutrition knowledge due to low literacy and language barriers [83]. For example, indigenous Fijians speak local dialects at home, however, the majority of health information is written in English. Thus, if an individual has limited literacy or English language abilities, they are at risk of not being able to access health information [84]. Therefore, it is imperative that nutrition knowledge be presented in a manner that makes the information accessible and easily understood by all individuals. In particular, practitioners should employ audio-visual communication methods such as radio and videos that are known to transcend barriers of literacy and language [85].

This study has limitations. This is a cross-sectional study that could not determine the cause-effect relationship. As this is a feasibility study, findings are limited to the study participants and cannot be generalised to the rest of the Fijian population. Because agri-businesses in Fiji are predominately self-employed families, we chose to capture dietary diversity at the household level. However, it is not uncommon for studies to assess dietary diversity at the individual level, which limits comparison with other studies. Furthermore, it is quite possible that the household dietary diversity scores do not reveal intra-household differences in food distribution. Although dietary diversity scores are criticised for not assessing serving sizes and quantified nutrient intake, studies have demonstrated that they are a low-cost method of assessing overall diet quality that is positively associated with micronutrient adequacy [55,56]. Over 70% of surveyed households only ate food prepared within the house in the 24-h food recall period demonstrating that data analysed in this sample may be representative of an ‘everyday’ diet. However, respondents self-reported dietary intake, which means data could be inaccurate due to under- or over-reporting. We are cognizant that survey administrators included some people in positions of power (i.e., Ministry of Agriculture extension officers) that could have biased participant reporting. However, existing rapport is conducive to participant recruitment in collectivist-based cultures and all local survey administrators lived and worked with the communities that participated in this study, which may have contributed to positive community engagement [49]. This was a pilot study undertaken during one season (harvest), which means we were unable to determine seasonal fluctuations in dietary diversity. The study sample was also small, which created large standard errors for some variables and lowered statistical power. Future research should seek to study dietary intake in greater detail and across different seasons with larger sample sizes. Dietary intake is mediated by social and cultural constructs, therefore a qualitative exploratory study would also be of benefit to gain understanding and context behind the findings of this study [86].

## 5. Conclusions

In indigenous food-producing households in rural Fiji, we observed a dietary pattern that included more traditional foods in comparison with the national average. Households with higher farm diversity or with more household occupants were significantly more likely to have higher dietary diversity. These findings demonstrate how important access to land and fishing grounds is to enable dietary diversity of poor households. Despite being involved in food production, consumption of empty-calorie foods in households was high indicating that rural populations are undergoing transition to a ‘modern’ diet. During nutrition transitions, dietary diversity initiatives must focus on developing culturally relevant programs that promote farm diversity and consumption of traditional diets that are low in sugar and processed foods and high in vegetables, fruits and lean protein.

## Figures and Tables

**Table 1 nutrients-11-01629-t001:** Descriptive statistics for respondents, households, farm and household dietary diversity (*n* = 161).

Variables	*n*	(%)	Mean ± SD
Respondent characteristics			
Gender			
Female	117	(72.7)	
Male	44	(27.3)	
Age (years)			
18–54	121	(75.2)	
≥55	40	(24.8)	
Education (years)			
≤12 (did not complete secondary school)	114	(70.8)	
≥13 (completed secondary school or higher)	47	(29.2)	
Employment			
Unemployed (caregiver)	110	(68.3)	
Employed	51	(31.7)	
Self-reported chronic health condition(s) (*n* = 180)			
Arthritis	19	(10.6)	
Asthma	11	(6.1)	
Back/-neck pain	63	(35.0)	
Cancer	1	(0.6)	
Depression/anxiety	12	(6.7)	
Diabetes	18	(10.0)	
Heart disease	3	(1.7)	
High blood pressure	50	(27.8)	
Kidney disease	2	(1.1)	
Stroke	1	(0.6)	
Household characteristics			
Gross annual household income (FJ$)			
≤5000	51	(32.3)	
≥5001	107	(67.7)	
Primary source of household income			
Self-employed smallholder farm	124	(77.0)	
Other (includes other small business)	18	(11.2)	
Private sector	11	(6.8)	
Public sector	5	(3.1)	
Remittance	3	(1.9)	
Household occupants			5.0 ± 2.3
1–5	98	(60.9)	
≥6	63	(39.1)	
Children 0–5-years-old living in household			1.0 ± 1.2
0–2	67	(41.6)	
≥3	94	(58.4)	
Food purchase frequency			
≥2/week	126	(78.3)	
≤1/week	35	(21.7)	
Farm diversity			
Farm status			
Subsistence	39	(24.2)	
Semi-commercial	118	(73.3)	
Commercial	4	(2.5)	
Crop Biodiversity Index			7.1 ± 5.1
Low (1–7)	109	(67.7)	
High (8–28)	52	(32.3)	
Livestock Biodiversity Index			0.9 ± 1.2
Low (0)	87	(54.0)	
High (1–5)	74	(46.0)	
Farm Diversity			7.9 ± 5.2
Low (1–7)	100	(62.1)	
High (8–28)	61	(37.9)	
Household dietary diversity			
Household Dietary Diversity Score (between 0–12)			7.8 ± 1.5
Low (1–6)	31	(19.3)	
Medium (7–9)	107	(66.5)	
High (10–12)	23	(14.3)	
Minimum Acceptable Diet Score (between 0–7)			4.3 ± 1.2
Low (1–3)	38	(23.6)	
Medium (4–5)	92	(57.1)	
High (6–7)	31	(19.3)	

SD = Standard deviation.

**Table 2 nutrients-11-01629-t002:** Frequency of household dietary diversity and farm diversity by food groups (*n* = 161).

Food Groups ^1^	Examples of Foods	Household Dietary Diversity *n* (%)	Farm Diversity ^2^ *n* (%)
High-sugar food/drink	Tea with sugar, sweets, cake, custard pie, lollies	158 (98)	0 (0)
Refined grains	White rice; white wheat-based bread, noodles, and roti	156 (97)	0 (0)
White roots and tubers	*Cassava*, *taro*, plantains (cooking bananas), white yams, white potato	151 (94)	156 (97)
Flavorings/other drinks	Lemon-leaf tea, flavorings, salt, ginger, garlic, chilies, spices, herbs	146 (91)	19 (12)
Oils and fats	Vegetable oil, ghee, butter, coconut cream	132 (82)	0 (0)
Dark green leafy vegetables	*Bele*, taro leaves (*rourou*), *cassava* leaves, wild spinach, english cabbage, chinese cabbage	124 (77)	105 (65)
Other vegetables ^3^	Tomato, cucumber, okra, long-beans, french-beans, cowpeas, eggplant, corn, green capsicum, zucchini, onion	117 (73)	110 (68)
Fish and seafood	Fresh fish, tinned fish, freshwater mussels, prawns, eel, octopus, crab	78 (48)	59 (37)
Meat	Chicken, pork, beef, mutton	64 (40)	31 (19)
Other fruits	Ripe banana, apple, watermelon, citrus (lemon, lime), pineapple, soursop, passionfruit	63 (39)	54 (34)
Vegetables, orange-fleshed	Pumpkin, carrot, sweet potato	57 (35)	59 (37)
Dried legumes and nuts ^4^	Dhal, yellow split-peas, dried green peas, peanuts, peanut butter	51 (32)	2 (1)
Dairy products	Powered-milk, long-life milk	52 (32)	0 (0)
Eggs	Chicken eggs	40 (25)	10 (6)
Fruits, orange-fleshed	Ripe papaya, ripe mango	37 (23)	31 (19)
Organ meat	Liver, kidney, heart	25 (16)	0 (0)
Other crops	Sugarcane, kava (*yaqona*), tobacco	Not collected	27 (17)

^1^ Household Dietary Diversity Score; ^2^ Farm diversity = food crops grown/-foraged, livestock bred/-hunted and seafood fished from ocean/-river; ^3^ Most commonly reported Other Vegetable was onion; ^4^ Refers to dried legumes as fresh beans and peas were included in Other Vegetables.

**Table 3 nutrients-11-01629-t003:** Percent of food groups consumed by households by dietary diversity tertile (*n* = 161).

Food Groups ^1^	Low ^2^ (*n* = 38; 24%)	Medium ^3^ (*n* = 92; 57%)	High ^4^ (*n* = 31; 19%)
Carbohydrates	100	100	100
Flesh meat	63	78	97
Vitamin-A rich fruits and vegetables	53	95	97
Other fruits and vegetables	34	93	97
Legumes and nuts	8	29	77
Eggs	8	26	77
Dairy	8	14	71

^1^ Minimum Acceptable Diet; ^2^ 1–3 food groups; ^3^ 4–5 food groups; ^4^ 6–7 food groups.

**Table 4 nutrients-11-01629-t004:** Univariate associations between socioeconomic characteristics and dietary diversity (*n* = 161).

	Low (1–3) * Minimum Acceptable Diet Score			Medium (4–5) * Minimum Acceptable Diet Score		
	β	(95% CI)	*p*	β	(95% CI)	*p*
Gender (Male)						
Female	1.53	(0.53–4.44)	0.429	1.28	(0.53–3.08)	0.588
Age (18–54-years)						
≥55–years	1.33	(0.46–3.82)	0.600	0.80	(0.31–2.05)	0.641
Education (≤12-years)						
≥13–years	2.02	(0.73–5.58)	0.174	2.05	(0.87–4.80)	0.099
Employment (Employed)						
Unemployed	7.39	(2.39–22.78)	0.001	3.33	(1.44–7.74)	0.005
Income (≤FJ$5000)						
≥FJ$5,001	0.52	(0.18–1.57)	0.248	1.06	(0.44–2.55)	0.892
Household Occupants (1–5)						
≥6	0.14	(0.05–0.43)	0.000	0.43	(0.18–0.98)	0.044
Children 0–5-years–old (0–2)						
≥3	1.09	(0.42–2.83)	0.855	0.75	(0.33–1.70)	0.485
Food purchase (≤1/week)						
≥2/week	0.57	(0.13–2.50)	0.457	0.27	(0.08–0.97)	0.045
Farm diversity (High)						
Low	9.14	(2.81–29.76)	0.000	1.97	(0.86–4.49)	0.108

* Reference category: Minimum Acceptable Diet = High (6–7); CI = confidence interval; *p* = <0.05.

**Table 5 nutrients-11-01629-t005:** Predictors of dietary diversity from adjusted multinomial logistic regression model (*n* = 161).

	Low (1–3) * Minimum Acceptable Diet Score			Medium (4–5) * Minimum Acceptable Diet Score		
	β	(95% CI)	*p*	β	(95% CI)	*p*
Employment (Employed)						
Unemployed	3.69	(1.02–13.44)	0.047	3.22	(1.24–8.37)	0.017
Household Occupants (1–5)						
≥6	0.14	(0.04–0.46)	0.001	0.36	(0.15–0.88)	0.024
Food purchase (≤1/week)						
≥2/week	0.37	(0.75–1.85)	0.228	0.21	(0.06–0.81)	0.023
Farm Diversity (High)						
Low	5.06	(1.34–19.13)	0.017	1.24	(0.48–3.21)	0.661

* Reference category: Minimum Acceptable Diet = High (6–7); CI = confidence interval; *p* = <0.05.

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
