# Peer review of "Predictors of Dietary Diversity of Indigenous Food-Producing Households in Rural Fiji"

_nutrients, 2019, doi:10.3390/nu11071629_

Round 1

Reviewer 1 Report

This an interesting paper and makes an important contribution to the literature. I have a only a few comments.

Introduction is overly long long.

Line 37. Use of quotations around phrase suggests that all three references cited used exactly the same. Hard to believe.

Lines 72-74. Example can be skipped since it is actually not relevant.

Section 2.3. Sample villages were randomly selected and hence it is unclear who representative they are.  Also not clear if all households in each of the villages selected were included or if some refused to participate, or if there was a some mechanism to select household within each village. These issues need to be clarified.

Section 2.4. Survey administrators included some people in positions of power, and hence their presence could have biased the information participants reported. This is an important potential limitation and should be discussed.

Reviewer 2 Report

Summary:

Interesting subject area and well-written and presented. However, I have some design and methodological concerns; the dichotomising of continuous data and a lack of justification on the cut-off points used; and of the methodology of 24hr recall. Food-producing households and thus farming diversity requires a study design which includes a broader study time period that spans across seasonal variation, this I feel is vitally important for this study’s aim and is lacking in the current study design.

I feel this manuscript also requires more depth of discussion and focus on the population in this study, i.e. medium diversity, and the limitations of the study.

Specific comments:

Abstract:

I would include the predictors of your medium dietary diversity (i.e. unemployment) as this is the majority of your population before you state the low dietary diversity predictors.

Introduction:

line60-62 – I suggest splitting the references so that your statement on ‘farming households’ is referenced and then you discuss the other studies described in the later part of this sentence

Methods:

Why only conducted over 1 one month period?

Line 101:  No description of sample size and power calculation only the selection of 8 villages targeted for recruitment

Line 120: short duration of the study and impact on seasonal variation will be important in discussion

Section 2.5: why dichotomised at 54yrs ? Why at 5000 Fijian dollars? Why at 6 household occupants? What is the average household size? What is the justification for dichotomising continuous data? 

food purchase frequency? Food spending would eb more useful unless you have justification of why frequency is important?

Section 2.7: what 24hr recall method was used i.e. prompts/cues to aid recall such as the 5step method (AMPM)?

Results:

Line 189: 5 villages so what happened to the other 3 that were targeted at recruitment? Exclusions?

Table 1: dichotomised data really doesn't show me anything, no justification for cut-offs

Section 3.4: I think this description of the data would read better if you describe as ‘Households with low DD are more likely than high diversity to be unemployed or low farm diversity but less likely than those with high dietary diversity to have households with more than 6 occupants.

Discussion:

Line 264: you state results regarding food purchasing frequency but I find no discussion about this finding

Lines 290-296: I think you could re-structure this discussion

Line 305: ‘Despite apparent high levels of dietary diversity...’ – your population was predominantly medium and low dietary diversity. Furthermore, in this paragraph need to expand on discussion that high diet diversity (method of counting) is not synonymous with good quality diet

Line 317: discuss why these crops may not be being grown by the food producers

Line 322: need to discuss the seasonal variation and effect on crop diversity and how this could influence the results of your study. What hypotheses do you have on this?

Line 358: you only mention seasonal variation with no discussion. Also, small sample size – how did this compare to power calculation?

I would also highlight the important of qualitative exploration study to gain understanding and context behind this survey.

Round 2

Reviewer 2 Report

Thank you for revising your manuscript according to both reviewers comments and suggestions.